# Reaction Time and Visual Memory in Connection with Alcohol Use in Schizophrenia and Schizoaffective Disorder

**DOI:** 10.3390/brainsci11060688

**Published:** 2021-05-23

**Authors:** Atiqul Haq Mazumder, Jennifer Barnett, Nina Lindberg, Minna Torniainen-Holm, Markku Lähteenvuo, Kaisla Lahdensuo, Martta Kerkelä, Jarmo Hietala, Erkki Tapio Isometsä, Olli Kampman, Tuula Kieseppä, Tuomas Jukuri, Katja Häkkinen, Erik Cederlöf, Willehard Haaki, Risto Kajanne, Asko Wegelius, Teemu Männynsalo, Jussi Niemi-Pynttäri, Kimmo Suokas, Jouko Lönnqvist, Solja Niemelä, Jari Tiihonen, Tiina Paunio, Aarno Palotie, Jaana Suvisaari, Juha Veijola

**Affiliations:** 1Department of Psychiatry, University of Oulu, 90014 Oulu, Finland; Martta.Kerkela@oulu.fi (M.K.); Tuomas.Jukuri@oulu.fi (T.J.); juha.veijola@oulu.fi (J.V.); 2Cambridge Cognition, University of Cambridge, Cambridge CB25 9TU, UK; jhb32@cam.ac.uk; 3Department of Psychiatry, Helsinki University Hospital, University of Helsinki, 00029 Helsinki, Finland; nina.lindberg@helsinki.fi (N.L.); erkki.isometsa@helsinki.fi (E.I.); tuula.kieseppa@helsinki.fi (T.K.); asko.wegelius@fimnet.fi (A.W.); tiina.paunio@helsinki.fi (T.P.); 4Mental Health Unit, Finnish Institute for Health and Welfare (THL), 00271 Helsinki, Finland; minna.torniainen-holm@thl.fi (M.T.-H.); erik.cederlof@thl.fi (E.C.); jouko.lonnqvist@thl.fi (J.L.); jaana.suvisaari@thl.fi (J.S.); 5Department of Forensic Psychiatry, Niuvanniemi Hospital, University of Eastern Finland, 70240 Kuopio, Finland; Markku.Lahteenvuo@niuva.fi (M.L.); Katja.Hakkinen@niuva.fi (K.H.); jari.tiihonen@ki.se (J.T.); 6Institute for Molecular Medicine Finland (FIMM), University of Helsinki, 00014 Helsinki, Finland; kaisla.lahdensuo@icloud.com (K.L.); hawker@utu.fi (W.H.); risto.kajanne@helsinki.fi (R.K.); teemu.mannynsalo@hel.fi (T.M.); jussi.niemi-pynttari@hel.fi (J.N.-P.); kimmo.suokas@tuni.fi (K.S.); aarno.palotie@helsinki.fi (A.P.); 7Mehiläinen, Pohjoinen Hesperiankatu 17 C, 00260 Helsinki, Finland; 8Department of Psychiatry, University of Turku, 20014 Turku, Finland; jarmo.Hietala@tyks.fi (J.H.); solnie@utu.fi (S.N.); 9Department of Psychiatry, Turku University Hospital, 20521 Turku, Finland; 10Faculty of Medicine and Health Technology, Tampere University, 33014 Tampere, Finland; olli.kampman@tuni.fi; 11Department of Psychiatry, Pirkanmaa Hospital District, 33521 Tampere, Finland; 12Social Services and Health Care Sector, City of Helsinki, 00099 Helsinki, Finland; 13Department of Psychiatry, University of Helsinki, 00014 Helsinki, Finland; 14Department of Clinical Neuroscience, Karolinska Institute, 17177 Stockholm, Sweden; 15Center for Psychiatry Research, Stockholm City Council, 11364 Stockholm, Sweden; 16Stanley Center for Psychiatric Research, The Broad Institute of MIT (Massachusetts Institute of Technology) and Harvard, Cambridge, MA 02142, USA; 17Analytical and Translational Genetics Unit, Massachusetts General Hospital, Boston, MA 02114, USA; 18Department of Psychiatry, Oulu University Hospital, 90220 Oulu, Finland

**Keywords:** cognition, visual memory, reaction time, alcohol, schizophrenia, schizoaffective disorder

## Abstract

The purpose of this study was to explore the association between cognition and hazardous drinking and alcohol use disorder in schizophrenia and schizoaffective disorder. Cognition is more or less compromised in schizophrenia, and schizoaffective disorder and alcohol use might aggravate this phenomenon. The study population included 3362 individuals from Finland with diagnoses of schizophrenia or schizoaffective disorder. Hazardous drinking was screened with the AUDIT-C (Alcohol Use Disorders Identification Test for Consumption) screening tool. Alcohol use disorder (AUD) diagnoses were obtained from national registrar data. Participants performed two computerized tasks from the Cambridge Automated Neuropsychological Test Battery (CANTAB) on a tablet computer: The Five-Choice Serial Reaction Time Task (5-CSRTT) or the reaction time (RT) test and the Paired Associative Learning (PAL) test. The association between alcohol use and the RT and PAL tests was analyzed with log-linear regression and logistic regression, respectively. After adjustment for age, education, housing status, and the age at which the respondents had their first psychotic episodes, hazardous drinking was associated with a lower median RT in females and less variable RT in males, while AUD was associated with a poorer PAL test performance in terms of the total errors adjusted scores (TEASs) in females. Our findings of positive associations between alcohol and cognition in schizophrenia and schizoaffective disorder are unique.

## 1. Introduction

Cognitive dysfunction is a persistent, disabling hallmark of schizophrenia found to be present in approximately 75% of schizophrenia patients [1]. It is associated with the severity of illness in schizophrenia [2]. Schizophrenia patients with comorbid alcohol use disorder (AUD) demonstrate a marked impairment in multiple cognitive domains [3,4,5,6,7]. The most affected cognitive domains are sustained attention, concept formation, and non-verbal cognitive flexibility [8,9].

Individuals with schizophrenia have a three-times greater risk of heavy alcohol use than the general population [10]. AUD is the second most common comorbidity in patients with schizophrenia, after nicotine dependence [11]. A systematic review and meta-analysis of 123 research papers published between 1990 and 2017 revealed a lifetime prevalence of AUD of 24.3% among subjects with schizophrenia [12].

Approximately 90% of people who drink excessively would not be expected to meet the clinical diagnostic criteria for having AUD [13]; however, they could screen positive for hazardous drinking or binge (heavy episodic) drinking.

Hazardous drinking is a pattern of alcohol consumption that increases the risk of harmful consequences for the user or others. Hazardous drinking patterns are of public health significance, irrespective of the absence of any current disorder in the individual alcohol user [14,15,16,17].

In the general population, males drink more alcohol than females [18]. However, in recent years, the male-to-female ratios for alcohol use, problematic alcohol use, and alcohol-related harm have declined considerably [19,20].

Studies in general populations suggest that mild-to-moderate alcohol drinking might not reduce cognition [21,22,23,24]. In schizophrenia patients, the association between cognition and different drinking patterns has not been fully studied yet.

The main aim of the present study was to explore the association between reaction time and visual memory with different drinking patterns in persons with schizophrenia and schizoaffective disorder diagnoses.

The specific research aims were to study the following:The association between hazardous drinking and reaction time and visual memory in persons with schizophrenia and schizoaffective disorder.The association between alcohol use disorder and reaction time and visual memory in persons with schizophrenia and schizoaffective disorder.

## 2. Materials and Methods

### 2.1. Participants

The participants of this study were part of the study population of SUPER (*Suomalainen psykoosisairauksien perinnöllisyysmekanismien tutkimus* (“Finnish Study of the Hereditary Mechanisms behind Psychotic Illnesses”)), which is part of the international Stanley Global Neuropsychiatric Genomics Initiative, USA. SUPER collected data during the period 2016–2019 from people with a lifetime diagnosis of psychosis in Finland to identify gene loci and gene variations predisposing patients to psychotic illnesses and comorbid diseases. Voluntary subjects with a diagnosis of schizophrenia spectrum psychotic disorder, bipolar I disorder, or major depressive disorder with psychotic features were recruited from psychiatric inpatient and outpatient departments, general healthcare centers, and supported housings. Participants were identified through local healthcare centers throughout the country from all levels of healthcare to ensure inclusive sampling. Subjects were also recruited via advertisements in local newspapers.

Out of the original sample of 10,555 participants, 6769 had clinical diagnoses of schizophrenia and schizoaffective disorder. Among those, 228 had missing information on alcohol use or education. Of the remaining 6541 participants, 1435 did not complete the cognitive tests or had duplicate test results. Finally, 3362 participants were included after excluding 1744 who did not live independently (living in supported housing, a hospital, or an unknown residence), as alcohol use is restricted in these places (Figure 1).

### 2.2. Schizophrenia and Schizoaffective Disorder Diagnoses

The diagnoses of schizophrenia and schizoaffective disorder were obtained from the Care Register for Health Care (CRHC) of the Finnish Institute for Health and Welfare. Upon fulfilling the diagnostic criteria, clinical diagnoses were made by clinicians, mostly in specialized care by psychiatrists, and the validity of the CRHC-based psychosis diagnoses was found to be good [25]. In Finland, the International Classification of Disease (ICD) system is used in psychiatric diagnoses. In this study, schizophrenia diagnoses included codes 295, according to ICD-8 and ICD-9, and F20, according to ICD-10, while schizoaffective disorder diagnoses included codes 295.7, according to ICD-8 and ICD-9, and F25, according to ICD-10. ICD-8 was used from 1968 to 1986 and ICD-9 from 1987 to 1995, while ICD-10 has been used since 1996 in Finland.

### 2.3. Hazardous Drinking Screening

Hazardous drinking was screened using the AUDIT-C questionnaire to assess an individual’s alcohol consumption frequency (“How often do you have a drink containing alcohol?”), quantity (“How many drinks containing alcohol do you have on a typical day when you are drinking?”), and bingeing (“How often do you have six or more drinks on one occasion?”). AUDIT-C is derived from the hazardous alcohol use domain of the Alcohol Use Disorders Identification Test (AUDIT) questionnaire [26]. It has three questions and is scored on a scale of 0 to 12. Each AUDIT-C question has five answer choices valued from zero to four points.

Cutoff scores for hazardous drinking vary considerably [26,27]. In the present study, we used the cutoff scores recommended by Finnish National Guidelines: a score of 6 or more in males and 5 or more in females [28].

### 2.4. Alcohol Use Disorder Diagnoses

The diagnoses of alcohol use disorder were obtained from the CRHC data according to codes ICD-8 291 and 303; ICD-9 291, 3030, and 3050A; and ICD-10 F10 for the period from 1969 to 2018.

### 2.5. Cognitive Measures

Processing speed and visual learning are the two core cognitive domains that are compromised in schizophrenia [29]; thus, we selected the Five-Choice Serial Reaction Time Task (5-CRTT) and the Paired Associative Learning (PAL) task from the Cambridge Neuropsychological Test Automated Battery (CANTAB) for schizophrenia for the assessment of reaction time (RT) and visual memory, respectively.

These tasks were chosen to produce relevant information on cognition in psychotic disorders in the very restricted assessment schedule. The instructions for both tests were translated into Finnish. The CANTAB tests were performed before venipuncture in order to avoid malfunction of the arm due to pain or bandaging. The study nurses were given standardized instructions on how to guide the study subjects in performing the CANTAB test beforehand.

In the RT test, we used two continuous measurements: the median of the five-choice reaction time and the standard deviation (SD) of the five-choice reaction time. The median of the five-choice reaction time is the median duration between the onset of the stimulus and the release of the button. The standard deviation of the five-choice reaction time is the standard deviation of the time taken to touch the stimulus after the button has been released. Both variables were calculated for correct, assessed trials where the stimulus could appear in any of the five locations.

In the PAL test, we assessed visual memory using the primary outcome variable of “total errors adjusted”. Here, we assessed two different dichotomized variables using data from the Northern Finland Birth Cohort 1966 (NFBC 1966) as reference data [30]. The NFBC 1966 consists of all children born with an expected date in the year 1966. The data used in this study consisted of a 46-year follow-up, when cohort members took the PAL test during clinical examination (*N* = 5608). For the first trial memory score, the 15th percentile (a score of 17 or more) was used as a cutoff for good performance in the PAL test in the recent study, meaning the SUPER population showed better performance than 15% of the NFBC 1966 study population. Regarding the scores for total errors adjusted for the NFBC 1966, the 50th percentile (an error score of 10 or less) was used as a cutoff for good performance in the PAL test in a recent study, meaning the SUPER population exhibited better error scores than 50% of the NFBC 1966 study population. The distribution of the PAL test is problematic, with multiple peaks. It does not follow any known distribution and thus we could not use a continuous distribution of PAL test variables. The first trial memory score shows how many patterns the participant correctly placed on the first attempt at each problem, while the total errors adjusted score reflects how quickly the participant learned when the participant was given multiple attempts at each problem.

### 2.6. Confounding Factors

Age, education [31], living without a spouse [32], and the age of a person at their first psychotic episode [4] have effects on cognitive functioning, and thus we considered them as the confounding variables in this study.

#### 2.6.1. Age

Cognition is negatively associated with increased age in healthy populations [33] and debatably in alcohol users [34]. The age of the participants was calculated using the participation date and year of birth of the participant. Age was used as continuous variable.

#### 2.6.2. Education

Education is strongly associated with cognitive performance [35]. The question and possible answers addressing the education of the participants were as follows: “What is your basic education?” (1 = less than primary school, 2 = matriculation examination, 3 = middle school, 4 = partial general upper secondary school or general upper secondary education certificate, 5 = partial middle school or primary school less than nine years, 6 = primary school, 7 = four-year elementary school). During the analysis, we combined classes 1, 3, 4, 5, 6, and 7 as “No matriculation examination” versus class 2 (“Matriculation examination”).

#### 2.6.3. Household Pattern

Household patterns, especially living without a spouse, might affect cognition [32,36], and thus we considered household patterns as a confounder.

#### 2.6.4. Age of First Psychotic Episode

We used the age of participants at their first psychotic episode as a marker of the severity of illness. Research shows that the earlier the development of schizophrenia and schizoaffective disorder, the more severe the illness in terms of the disease pattern and cognitive decline [37,38]. The data regarding the age of people at the time of their first psychotic episode were obtained from the CRHC and were used in this research as a continuous variable.

### 2.7. Statistical Methods

We evaluated the association between cognition and alcohol use by using four different cognition variables: the median and standard deviation of RT, PAL FTMS, and PAL with total errors adjusted. Alcohol use was measured with different variables: the dichotomous hazard drinking variable was derived from the AUDIT score and the dichotomous variable indicated if the study subject had an alcohol use disorder diagnosis. We assessed crude models and also adjusted models with age, education, household patterns, and age of participants at the time of their first psychosis episode. The association between the RT test and alcohol use was analyzed with log-linear regression and eβ with 95% confidence intervals (CI) are reported. The association between the PAL test and alcohol use was analyzed with logistic regression and odds ratios (ORs) with 95% CIs are reported.

All the analyses were conducted separately for males and females. Males and females showed differences in performing the selected cognitive tests [39,40,41]; additionally, males and females showed differences in alcohol use patterns [42].

## 3. Results

### 3.1. Background Factors and Alcohol Use Patterns

Of the participants, 51% were male and 49% were female. The mean age was 45 years for males and 46 years for females. One-third of the males and two-fifths of the females had the highest basic educational level of 12 years (matriculation). The mean age at the time of the first psychotic episode was 27 years for males and 28 for females. One-tenth of the males and one-fourth of the females were living with a spouse. Most of the participants were on psychotropic medication (Table 1).

Hazardous drinking and AUD were more common in males than in females. Every fourth male and every sixth female schizophrenia or schizoaffective disorder patient showed a positive screening for hazardous drinking. Approximately one-third of male and one-sixth of female schizophrenia or schizoaffective disorder patients had a lifetime diagnosis of AUD (Table 1).

Lower age was associated with hazardous drinking, both in males and females. In females, a lower age of onset and types of household were associated with hazardous drinking (see Appendix A).

The median RT was 435 ms (SD = 50 ms); the PAL median FTMS was 8 and the median total errors adjusted was 33 (see Appendix A).

The association between background factors and AUD with RT *p*-values is reported in the Appendix A. The association between background factors and alcohol use patterns with PAL scores is reported in the Appendix A.

The Cohen’s d measure of effect size is shown in the Appendix A. The effect sizes were in line with our results (significant results were found for male hazardous drinking RTI SD (*p* = 0.008; Cohen’s d = 0.22), female RTI median (*p* = 0.016; Cohen’s d = 0.31 (0.18–0.44)), and female PAL TEA alcohol use disorder (*p* = 0.048; Cohen’s d = 0.12)).

### 3.2. Association between Reaction Time and Visual Memory and Hazardous Drinking in Schizophrenia and Schizoaffective Disorder

After adjustment for age, education, household patterns, and age at the time of the first psychotic episode, hazardous drinking was associated—in females—with a lower median RT (OR = 0.97; 95% CI, 0.95–0.99) and in males with a less variable reaction time (Table 2). The association between hazardous drinking and RT scores is reported in the Appendix A.

### 3.3. Association between Reaction Time and Visual Memory and Alcohol Use Disorder in Schizophrenia and Schizoaffective Disorder

There was no significant difference in reaction times between male and female schizophrenia and schizoaffective disorder patients with or without a lifetime history of alcohol use disorder (Table 3).

Females with AUD performed more poorly in the PAL test than females without AUD (OR = 1.34; 95% CI, 1.02–1.93) (Table 3).

## 4. Discussion

### 4.1. Main Findings

Our findings did not support our assumption that problematic drinking was associated with impaired cognitive function in schizophrenia and schizoaffective disorder. Rather, some positive associations were found between hazardous drinking and reaction time scores, both in males and females.

### 4.2. Comparison with Other Studies

There are hardly any studies investigating the association of different alcohol use patterns in schizophrenia and schizoaffective disorder patients; thus, it is difficult to compare our findings with other studies.

Most studies investigating the cognitive impact of alcohol in schizophrenia patients with comorbid AUD revealed a negative association between alcohol use and cognitive function [6,7,43,44,45,46]. On the contrary, some studies have suggested that AUD has no additive effects on cognitive impairment in schizophrenia patients [47,48,49] or has positive associations with social cognition [50,51,52,53], task-making, and the speed processing domains [54].

A meta-analysis of six studies published between 1996 and 2009 revealed that younger (<30 years) schizophrenia patients with comorbid alcohol use disorders have better cognition than schizophrenia patients without AUD. By contrast, older (>40 years) schizophrenia patients with comorbid alcohol use disorders have worse cognition than their non-comorbid counterparts [5]. A systematic review of research papers published between 1990 and 2012 revealed that cognition is more preserved in schizophrenia patients with comorbid substance/alcohol use disorders compared to schizophrenia patients without comorbidity [55]. It is likely that unmeasured confounding factors contributed to the discrepant findings in previous studies.

General population cross-sectional studies investigating the effects of alcohol on cognitive function have revealed that moderate-to-heavy drinking is associated with cognitive decline [56,57,58,59], while mild-to-moderate drinking is associated with either no effects on cognition [60,61] or cognitive enhancement [61,62,63,64,65].

Most cohort studies in the general population addressing the same issue have revealed a positive correlation between light/light-to-moderate alcohol use and cognitive function [23,24,66,67,68], whereas one cohort study found no association between light-to-moderate alcohol consumption and better or worse cognitive functions [69,70,71]. One study reported a positive association between moderate-to-heavy drinking and cognitive function [72]. By contrast, another cohort study revealed a negative association between heavy alcohol use and cognitive function in a normal population [73]. One cohort study found a dose-depended positive association for alcohol use and cognition compared to abstainers and former drinkers [74]. Another cohort study revealed significant cognitive impairment in low-functioning non-drinkers and light-to-moderate drinkers and high-functioning non-drinkers [75].

A brief review of 29 studies (2003–2013) revealed that acute alcohol use mostly impairs executive functioning in the normal population [76]. By contrast, a systematic review of 143 studies (1977–2011) revealed that light-to-moderate alcohol use does not impair cognition in young male and female individuals and reduces the risk of all forms of dementia and cognitive decline in older individuals [77]. Another systematic review of 28 reviews (2000–2017) revealed that light-to-moderate alcohol use in middle-to-late adulthood is associated with a decreased risk of cognitive impairment and dementia [22]. A meta-analysis of 27 cohort studies (2007–2018) revealed that moderate alcohol use improves cognition insignificantly among males and slightly among females compared to current non-drinkers [21]. Moderate alcohol use has been found to be associated with reduced amyloid-beta deposition in the human brain [78].

Study findings suggesting a positive association between alcohol use and cognition could be attributed to unmeasured or residual confounding factors [79,80], such as smoking [81], drink types [82], drinking patterns [83], personality [81,84], intelligence [70,85,86], educational attainment [87,88], potential abstainer errors [89,90,91,92], reverse causality bias [93], recall error [94], within-person temporal variation [95,96], ascertainment of diseases [97], and the sociability effect of alcohol [98]. Study findings suggesting a positive association between alcohol use and cognition could be attributed to poor motivation [99,100,101].

Some animal model studies found negative association between alcohol and cognition mediated by reductions in cell density in the cerebral cortex [102] and alterations of cholinergic interneurons in the Nucleus Accumbens [103]. Other animal model studies suggested positive association between alcohol and cognition thought to be mediated by enhancement of brain metabolite clearance [104] and activation of the vagus nerve [105], antioxidant system [106,107,108,109], and other biological processes [110,111,112]. However, none of these findings have thus far been confirmed to be causally important.

Epidemiological studies have suggested that the apparent beneficial effects of alcohol use might largely be non-causal [24,113,114,115,116]. More research is needed for further clarification [117].

### 4.3. Strengths

We were able to use a very large sample of schizophrenia and schizoaffective disorder patients to investigate the cognitive impact of different alcohol use patterns. We studied multiple alcohol use patterns in the same study population and used age, education, and age of onset of schizophrenia and schizoaffective disorder as potential confounding variables.

Although long-term antipsychotic medication might be associated with the additional loss of both gray and white matter in schizophrenia [118], in our study we did not confound antipsychotic medication because 98% of the study population was on antipsychotic medication; thus, it would not have made any statistically significant difference. First-generation antipsychotics have been found to be associated with an increased volume of the basal ganglia—namely, the globes pallidus—which might be reversed upon switching to second-generation antipsychotics and clozapine [119,120]. However, antipsychotic-induced brain volume loss has never been established to be associated with additional cognitive decline [121]. It was beyond the scope of this study to consider different types and different doses of psychotropic medication, because it would be extremely laborious to extract detailed data on medication and translate the prescription from Finnish to English. The use of drug combinations was a further problem. In the original self-reporting questionnaire of SUPER, only conditions using or not using antipsychotic medication were included.

We included all schizophrenia and schizoaffective disorder patients living independently and excluded those whose living circumstances might have affected their alcohol use.

### 4.4. Limitations

We used only two tests from CANTAB. Our study was cross-sectional, not longitudinal. We did not use information about the onset of alcohol use, any recent changes in drinking habits, or any previous history of abstinence. We also did not differentiate previous alcohol users from people who never used alcohol.

We did not use information about poly-substance use and smoking. We did not include daily smoking as a covariate because of multicollinearity; approximately three-quarters of those with hazardous alcohol use in this sample were also daily smokers.

### 4.5. What Is Already Known on This Subject?


Alcohol use disorder reduces cognition in schizophrenia and schizoaffective disorder patients.Mild alcohol use is not associated with impaired cognition in the normal population.


### 4.6. What Does This Study Add?

Hazardous drinking was not associated with cognitive deficits in schizophrenia and schizoaffective disorder.

## 5. Conclusions

Hazardous drinking was not associated with a cognitive decline in schizophrenia and schizoaffective disorder patients. Rather, some positive associations were found between hazardous drinking and cognition, which are unique to these psychiatric disorders but in line with the findings from general population studies. Further genetic epidemiological studies could help to resolve the correlation–causality–reverse causality debate regarding alcohol and cognition.

## Figures and Tables

**Figure 1 brainsci-11-00688-f001:**
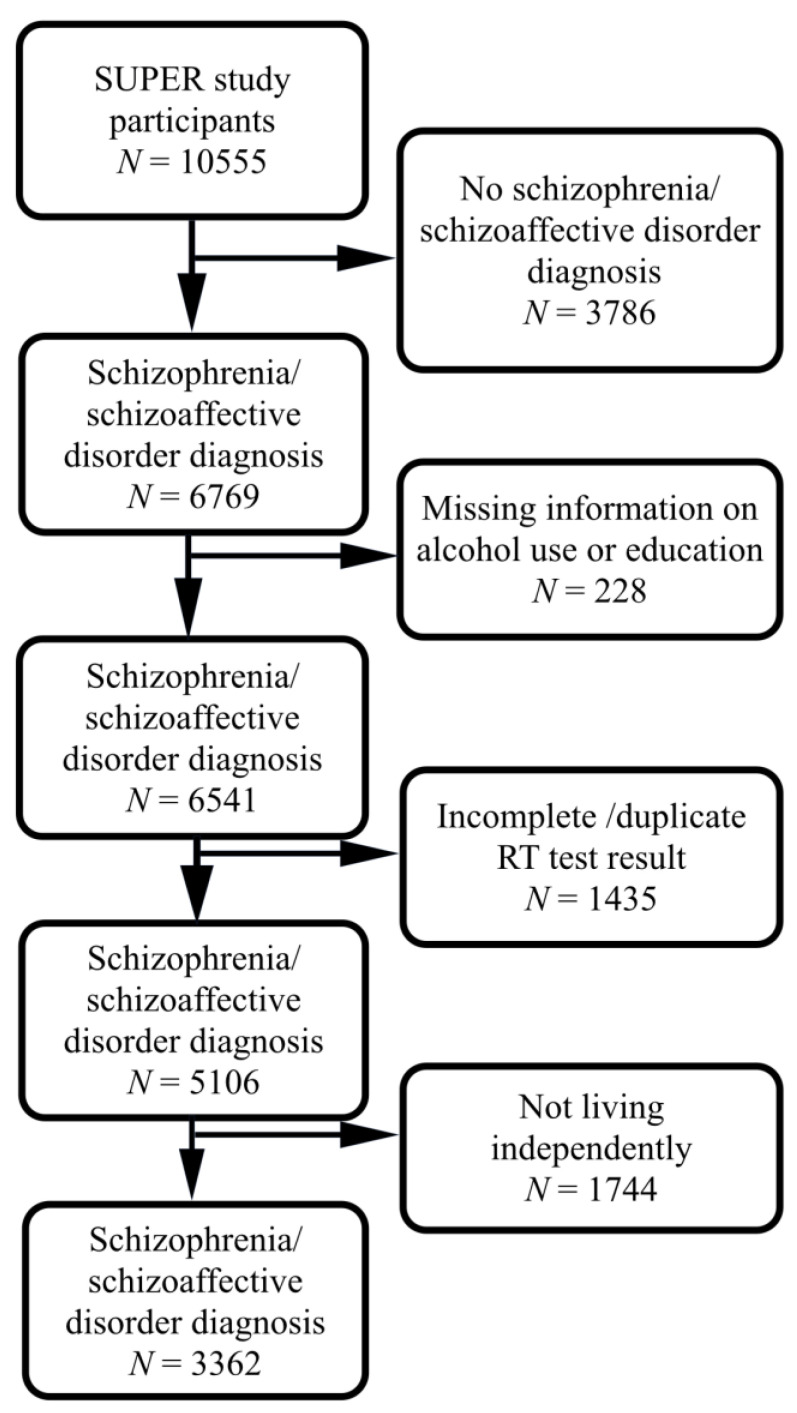
Flowchart showing the selected study population.

**Table 1 brainsci-11-00688-t001:** Background factors and alcohol use patterns in schizophrenia and schizoaffective disorder.

	Male	Female
	*N* = 1711	*N* = 1651
**Age (mean (SD))**	44.5 (12.8)	46.3 (13.2)
**Education**		
No matriculation examination (%)	1191 (69.6)	981 (59.4)
Matriculation examination (%)	520 (30.4)	670 (40.6)
**Age of first psychotic episode (mean (SD))**	26.84 (8.25)	28.01 (9.51)
**Household patterns**		
With spouse	200 (11.7)	441 (26.7)
Other	1511 (88.3)	1210 (73.3)
**Current psychotropic medications**		
No (%)	42 (2.45)	33 (2.00)
Yes (%)	1668 (97.49)	1614 (97.76)
Missing (%)	1 (0.06)	4 (0.24)
**Hazardous drinking ***		
No (%)	1276 (74.6)	1390 (84.2)
Yes (%)	435 (25.4)	261 (15.8)
**Alcohol use disorder**		
No (%)	1212 (70.8)	1395 (84.5)
Yes (%)	499 (29.2)	256 (15.5)

* AUDIT-C cutoff scores for hazardous drinking were ≥6 for males and ≥5 for females.

**Table 2 brainsci-11-00688-t002:** Association between RT and PAL tests and hazardous drinking in schizophrenia and schizoaffective disorder.

	Five-Choice Reaction Time *				Five-Choice Reaction Time *			
	Median				SD			
	**Crude**		**Adjusted ^a^**		**Crude**		**Adjusted ^a^**	
	**OR (95% CI)**	***p*-value**	**OR (95% CI)**	***p*-value**	**OR (95% CI)**	***p*-value**	**OR (95% CI)**	***p*-value**
**Male**								
Hazardous drinking	0.97 (0.95–0.99)	0.002	0.99 (0.97–1.01)	0.379	0.86 (0.81–0.91)	<001	0.93 (0.88–0.98)	0.008
**Female**								
Hazardous drinking	0.94 (0.92–0.97)	<0.001	0.97 (0.95–0.99)	0.016	0.89 (0.83–0.96)	0.002	0.99 (0.92–1.06)	0.760
	**PAL first trial memory score ****				**PAL total errors adjusted score ****			
	**Better performance than 15% of NFBC 1966 members**				**Higher error scores than 50% of NFBC 1966 members**			
	**Crude**		**Adjusted ^a^**		**Crude**		**Adjusted ^a^**	
	**OR (95% CI)**	***p*-value**	**OR (95% CI)**	***p*-value**	**OR (95% CI)**	***p*-value**	**OR (95% CI)**	***p*-value**
**Male**								
Hazardous drinking	1.76 (1.13–2.66)	0.001	1.24 (0.74–2.02)	0.399	0.56 (0.43–0.73)	<0.001	0.92 (0.67–1.28)	0.253
**Female**								
Hazardous drinking	1.58 (0.95–2.52)	0.064	1.02 (0.60–1.71)	0.893	0.58 (0.43–0.80)	<0.001	0.97 (0.75–1.34)	0.743

^a^ Adjusted for age, education, household patterns, and age at the time of the first psychotic episode. * Analyzed with log-linear regression. ** Analyzed with logistic regression. NFBC 1966 = Northern Finland Birth Cohort 1966, a population based birth cohort.

**Table 3 brainsci-11-00688-t003:** Association between RT and PAL tests and alcohol use disorder in schizophrenia and schizoaffective disorder.

	Five-Choice Reaction Time *				Five-Choice Reaction Time *			
	Median				SD			
	**Crude**		**Adjusted ^a^**		**Crude**		**Adjusted ^a^**	
	**OR (95% CI)**	***p*-value**	**OR (95% CI)**	***p*-value**	**OR (95% CI)**	***p*-value**	**OR (95% CI)**	***p*-value**
**Male**								
Alcohol use disorder	1.01 (0.99–1.02)	0.545	1.00 (0.99–1.02)	0.871	1.02 (0.96–1.08)	0.482	1.00 (0.95–1.06)	0.899
**Female**								
Alcohol use disorder	0.99 (0.96–1.01)	0.246	0.99 (0.96–1.01)	0.225	1.03 (0.96–1.11)	0.413	1.04 (0.97–1.11)	0.324
	**PAL first trial memory score ****				**PAL total errors adjusted score ****			
	**Better performance than 15% of NFBC 1966 members**				**Higher error scores than 50% of NFBC 1966 members**			
	**Crude**		**Adjusted**		**Crude**		**Adjusted**	
	**OR (95% CI)**	***p*-value**	**OR (95% CI)**	***p*-value**	**OR (95% CI)**	***p*-value**	**OR (95% CI)**	***p*-value**
**Male**								
Alcohol use disorder	0.71 (0.45–1.09)	0.131	0.83 (0.46–1.45)	0.543	1.36 (1.05–1.77)	0.021	1.18 (0.84–1.67)	0.358
**Female**								
Alcohol use disorder	0.53 (0.28–0.94)	0.043	0.62 (0.29–1.17)	0.102	1.57 (1.13–2.22)	0.009	1.34 (1.02–1.93)	0.048

^a^ Adjusted for age, education, household patterns, and age at the time of the first psychotic episode. * Analyzed with log-linear regression. ** Analyzed with logistic regression. NFBC 1966 = Northern Finland Birth Cohort 1966, a population based birth cohort.

## Data Availability

The raw data and materials used for this study are available upon request.

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
