# Peer review of "Reaction Time and Visual Memory in Connection with Alcohol Use in Schizophrenia and Schizoaffective Disorder"

_brainsci, 2021, doi:10.3390/brainsci11060688_

Round 1

Reviewer 1 Report

Thank you for the interesting research report. The manuscript explores the association of cognition with drinking and alcohol use disorder in schizophrenia and schizoaffective disorder, it is considered to be an important study that analyzed multiple factors using a large sample. And the topic addressed is interesting and deserves a constructive discussion. I would like to comment on the following 6 points.

recommendations for revision-minor

1. Line 124-131

Please describe the attributes of the diagnostician.

2. Line 147

A hyphen may be necessary: AUDIT"-"C

3. Line 183-184

Please add data to show the validity of this cutoff point.

4. Line 208

Even though there are previous studies, it is possible that age-based indicators alone are insufficient. The addition of clinical symptom assessment would allow for a more detailed study.

5. Line 236&416-418

It is desirable to consider or mention the effect of drug dosage.

6. Line 290-292

It is recommended to check the “high” or “low”.

I hope these comments will be helpful.

Reviewer 2 Report

The authors present an interesting set of finding regarding the associations of problematic alcohol use among persons with schizophrenia or schizoaffective disorder. Namely, that if anything they observed a positive association between more hazardous drinking and binge drinking (6 or more/event based on the AUDIT-C subscale) with cognitive performance. The data were derived from a large subset of participants (n = 3362) with the above diagnoses and living independently from the larger SUPER study in Finland. Analyses were conducted separately for males and females and by levels of reported drinking. Cognition was assessed using measures from 2 subtests of the CANTAB: Reaction time test (5-choice version; median and SD of RT) and the paired associative learning (PAL) test (first trial memory and total adjusted errors). Considering associations adjusted for demographics (age, education, first episode age), a few specific positive relations were observed. Faster RTs were predicted by hazardous drinking among females and by binge drinking monthly or less by both sexes. Better PAL memory performance related to binge drinking among males (weekly or more) and females (monthly or less). Diagnosis of an alcohol use disorder only related to one measure in females. The findings are nuanced, but failed to support an hypothesis of relations between problematic drinking and poorer cognitive performance in this sample. The findings are discussed well in the context of mixed relations reported in both the schizophrenia and more general literature. While adding a substantial set of data to this issue in the literature, I have a number of queries and comments for consideration regarding the study concept and analyses.

CONCEPTUAL AND DESIGN ITEMS

  1. In a couple of places the paper refers to assessing whether problematic alcohol use is associated with “additional cognitive impairment” in schizophrenia. However, the cognitive impairment is implicit – i.e., it’s not explicit what alcohol may be “adding” to in this sample and there is no healthy comparison. Typically, these are sensitive measures to psychosis. It may help, however, to characterize for readers what the performance levels on the CANTAB data represent relative to both normative data as well as other schizophrenia samples. The lack of healthy comparison or other samples could also be further considered as it’s unclear to what extent the current findings may be unique to the schizophrenia subsample.

  1. The introduction is very succinct and some elaboration would be useful to strengthen the rationale. For instance, it notes, “Schizophrenia patients with comorbid AUD demonstrate marked impairment in multiple cognitive domains… Most affected cognitive domains are sustained attention, concept formation and non-verbal cognitive flexibility [8]” (lines 62-64). The latter reference, however, specifically assessed executive functions. Regardless, given the widespread cognitive relations to schizophrenia and those pointed out here, there is little rationale provided for “The main aim of the present study was to explore the association of reaction time and visual memory.” Processing speed and memory are core areas of deficit in schizophrenia, but they aren’t explicitly acknowledged. If this is rather a convenience sample of data from these 2 domains from the SUPER study, that could be explained.

  1. Likewise, I believe the SUPER study only included the above 2 CANTAB measures, but this could be clarified. Without that understanding, it otherwise reads as though the authors arbitrarily selected them from the larger battery. The section describing the tasks, for example place emphasis on their relevance to Alzheimer’s disease and first-episode psychosis, whereas the current sample has more chronic psychosis.

  1. The SUPER study collected data from 2016-2019, but it is noted that diagnoses of SSDs and AUD were derived from records spanning back to 1960s and based on ICD8, 9, or 10 accordingly. This limitation might be considered further given both dated and different diagnositic versions. Medication use was noted. Was there any other verification of current symptoms or diagnoses?

  1. The discussion of main findings (4.1) focuses on the analysis of ‘crude’ associations, whereas a number of these became nonsignficant or had lower ORs once adjusting for demographic factors. It would be more appropriate to at least include the adjusted results as well as appreciation of the magnitudes of relations (see below.)

  1. It is unclear whether the authors think the findings may be unique to schizophrenia as mixed findings are discussed for general populations as well. Regardless, they might elaborate on why there may be these positive relations with mild levels of problematic use in schizophrenia. For example, might this support some form of self-medication?

STATISTICAL ISSUES

  1. The explanation for why thresholds for ‘good’ performance on the PAL were set at the 15th percentile for the first-trial memory and 50th percentile for errors is unclear. It seems to relate to how the parent SUPER population of participants compares to another sample. But what is the purpose of these cut-offs for the current study? This is important because the data are analyzed using logistic regression based on these cut-offs. Why not harness the full distribution of data to assess more continuous relations?

  1. Similarly why are separate analyses conducted for two level comparisons of binge drinking as opposed to analyses incorporating these the binge-drinking measure as a single predictor on its full 5-point scale?

  1. For the hazardous drinking analyses some ORs are significant, but the effect sizes are quite small, i.e., top portions of Tables 2 and 3 adjusted ORs range from .91-.98. ORs for the PAL scores are more slightly more robust, but none of them would exceed Cohen’s (and others’) qualitative ranges for deeming these as even “small” effect sizes. In the binge drinking analyses, only two of the male PAL FTMS ORs exceed 1.5 as ‘small’. Thus, the context of statistical significance afforded by the very large sample size needs to be considered and more interpretive weight given to the magnitude of these relations. It is not to say that very small effects are not meaningful, but the magnitude should be further considered.

  1. Likewise, the supplementary tables are evaluated based on p-values, but given large N it might be more apt to based interpretation on effect-size measures or at least mean differences.

  1. The models adjusted for age, education, and age of FEP. Household pattern also appears to be a highly significant correlate of performance and might also be considered as an important moderator.

  1. Related, as noted in the intro, alcohol use in SCZ is second to the high prevalence of smoking in this population. The discussion also points out smoking may be an important confound. Does the dataset include information on smoking? It may be interesting to assess these relations as nicotine can sometimes offset alcohol’s effects on cognition. Also, is it possible participants had smoked prior testing?

  1. Likewise notes personality and motivational factors; data available on current Sx?

  1. The manuscript repeatedly refers to the sample as comprising both “schizophrenia and schizoaffective disorder.” There have been debates about the distinction. Given the large N, it affords opportunity to assess whether these diagnoses moderate any of the results. Was this considered?

  1. The relations are small, but moreover appear to highlight that mild levels of problematic drinking are most responsive to the positive relations with cognition. This suggests overall nonlinear relations. It may be useful to have some visual representation of the findings – e.g., scatterplots or violin plots – to more fully visualize the set of relations. If limited in this regard, perhaps a supplementary table with the breakdown of scores across levels of drinking measures would help (like Supp Table 2, but by AUDIT categories).

  1. A final conclusion of what the study adds to the literature indicates, “Mild alcohol use is not associated with additional cognitive impairment in schizophrenia and schizoaffective disorder patients.” I appreciate this stance in light of the subtle positive relations that are opposite those hypothesized. Moreover, the discussion reviews many discrepancies in the literature regarding relations between drinking variables and cognition. It could add to the report to more directly assess the strength of the null finding – e.g., using Bayes factors to assess the probability of the null to the alternative hypotheses.

MINOR POINTS

  1. BD was defined by the AUDIT item for 6-plus drinks. The authors are constrained by this test from the SUPER study. However, this is considered high in many countries (e.g., 4 for females, 5 for males is often used by other measures) and is being evaluated based on a single item. This might be addressed as a potential limitation.
  2. The participants method section (p.3) end with two paragraphs seemingly pasted from a guide for authors noting that access numbers for public databases should be provided and that ethical approval should be provided. Likewise the results begin with an instructional paragraph. The endnotes do speak to ethics – it is unclear whether this relates to the full SUPER study or to the authors current secondary analyses. These items should be addressed prior acceptance.
  3. Given the large sample size, a majority who had not matriculated, and the more nuanced education data, why were they binarized? Again, there seems a theme in the analyses to truncate the categories from larger scales.
  4. The data shown in Table 2 are cluttered, Table 3 is slightly better. But these may display better in landscape orientation.

Round 2

Reviewer 2 Report

Thank you for your comprehensive responses, and clarifications and modifications to the manuscript. The intro rationale for the cognitive focus could be further clarified, but the issues are clarified elsewhere in the manuscript. Likewise, the effect sizes and relevant limitations are included for readers. Overall, the revisions adequately address my prior concerns and queries.  This large-scale study provides an interesting contribution to the field.